# Sublethal Radiation Affects Antigen Processing and Presentation Genes to Enhance Immunogenicity of Cancer Cells

**DOI:** 10.3390/ijms21072573

**Published:** 2020-04-07

**Authors:** Achamaporn Punnanitinont, Eric D. Kannisto, Junko Matsuzaki, Kunle Odunsi, Sai Yendamuri, Anurag K. Singh, Santosh K. Patnaik

**Affiliations:** 1Department of Thoracic Surgery, Roswell Park Comprehensive Cancer Center, Elm and Carlton Streets, Buffalo, NY 14263, USA; punnania@gmail.com (A.P.); eric.kannisto@roswellpark.org (E.D.K.); sai.yendamuri@roswellpark.org (S.Y.); 2Department of Gynecologic Oncology, and Center for Cancer Immunotherapy, Roswell Park Comprehensive Cancer Center, Elm and Carlton Streets, Buffalo, NY 14263, USA; junko.matsuzaki@roswellpark.org (J.M.); kunle.odunsi@roswellpark.org (K.O.); 3Department of Radiation Medicine, Roswell Park Comprehensive Cancer Center, Elm and Carlton Streets, Buffalo, NY 14263, USA

**Keywords:** antigen presentation, cancer cell line, gene expression, lung cancer, head and neck cancer, radiation, tumor antigen

## Abstract

While immunotherapy in cancer is designed to stimulate effector T cell response, tumor-associated antigens have to be presented on malignant cells at a sufficient level for recognition of cancer by T cells. Recent studies suggest that radiotherapy enhances the anti-cancer immune response and also improves the efficacy of immunotherapy. To understand the molecular basis of such observations, we examined the effect of ionizing X-rays on tumor antigens and their presentation in a set of nine human cell lines representing cancers of the esophagus, lung, and head and neck. A single dose of 7.5 or 15 Gy radiation enhanced the New York esophageal squamous cell carcinoma 1 (NY-ESO-1) tumor-antigen-mediated recognition of cancer cells by NY-ESO-1-specific CD8^+^ T cells. Irradiation led to significant enlargement of live cells after four days, and microscopy and flow cytometry revealed multinucleation and polyploidy in the cells because of dysregulated mitosis, which was also revealed in RNA-sequencing-based transcriptome profiles of cells. Transcriptome analyses also showed that while radiation had no universal effect on genes encoding tumor antigens, it upregulated the expression of numerous genes involved in antigen processing and presentation pathways in all cell lines. This effect may explain the immunostimulatory role of cancer radiotherapy.

## 1. Introduction

Radiotherapy can be tumoricidal at doses that are not directly lethal to cancer cells. One of the reasons for this is the elevated anti-tumor immune response that radiation can trigger at a sublethal dose. This benefit of radiotherapy has been known since 1979, when Stone et al. demonstrated that tumor radiocurability in mice was dependent on host immune capability [1]. With the advent of immunotherapy in recent years, the immune benefit of sublethal radiation is being put to use by utilizing radiotherapy as an adjuvant to boost responsiveness to immunotherapy [2], and multiple clinical trials on combination immunotherapy and radiotherapy are being conducted for a wide variety of malignancies [3,4,5], including non-small-cell lung cancer (NSCLC) [6] and squamous cell carcinomas of head and neck (HNSCC) [7]. The immune effect of radiotherapy results from the effects of radiation on both cancer cells and non-cancer stromal cells in the tumor microenvironment, including fibroblasts, immune cells, and cells of the vasculature [8,9,10,11]. In 1971, Natale et al. were among the first to show that sublethal radiation renders cancer cells more immunogenic [12]. The recent enthusiasm on the synergy of radiotherapy and immunotherapy has renewed interest in the mechanisms by which sublethal radiation enhances cancer immunogenicity [13]. We sought to investigate the universality of such mechanisms using a panel of human cancer cell lines.

## 2. Results

### 2.1. Irradiation of Cancer Cells Can Enhance Their Recognition by CD8^+^ T Cells

NY-ESO-1 (New York esophageal squamous cell carcinoma 1), also known as cancer/testis antigen 1B (CTAG1B), is a tumor-associated antigen that is expressed in a variety of cancers, including non-small-cell lung and head and neck carcinomas. This antigen can elicit both humoral and cellular immune responses in cancer patients [14]. We previously isolated from human patients cytotoxic CD8^+^ T cell clones that are NY-ESO-1-specific and restricted to the HLA-A*02 type of class I major histocompatibility complex (MHC) [15]. The NCI-H522 human lung adenocarcinoma cell line (H522) is of the HLA-A*02 type and expresses the *CTAG1B* gene, although at a low level compared to multiple other NSCLC cell lines [16]. Binding of MHC-I-presented NY-ESO-1 on H522 cell surface to NY-ESO-1-specific T cell receptors activates the T cells, which then secrete interferon γ (IFNγ). This permits quantification of cancer cell recognition with an enzyme-linked immune absorbent spot (ELIspot) assay directed against IFNγ. H522 cells do not produce IFNγ.

To examine whether radiation enhances the recognition of cancer cells by CD8^+^ T cells, we co-cultured H522 cells with our NY-ESO-1-specific CD8^+^ T cells at 5:1 ratio for 24 h. In line with our previous observations in human A498 renal carcinoma cells [17], irradiation of H522 with a single 7.5 Gy dose of X-rays three days prior to co-culture increased their T-cell recognition 1.4-fold (standard t test *p* = 0.02; Figure 1A). With a 15 Gy dose, the increase was 1.6-fold, although the difference in effects of the two doses was not statistically significant (*p* = 0.11). Similar observations were obtained in a replicated experiment, and in an experiment using the HLA-A*02^+^ human OE19 esophageal adenocarcinoma cell line (Figure 1B). *NY-ESO-1* RNA levels in the H522 and OE19 cell lines are similar [18].

### 2.2. Cell Surface Proteins of Tumor Antigens May Not Be Increased by Radiation Treatment of Cancer Cells

Having observed radiation-mediated enhancement of NY-ESO-1 cancer cell antigen presentation to CD8^+^ T cells with cell lines of three different cancers—esophagus, lung (Figure 1), and kidney (A498 cell line) [17]—we sought to understand the molecular basis of this phenomenon using a panel of three HNSCC and five NSCLC human cell lines (Table 1). Radiation therapy is an important mode of treatment for both HNSCC and NSCLC. The cell doubling time of the eight selected cell lines varied from about 22 to 96 h. Their radiation sensitivity, as measured by clonogenic survival fraction at 8 Gy (SF2), varied about 2-fold from 0.43 to 0.72. For comparison, among the 54 non-lymphoid human cancer cell lines of the NCI-60 panel representing eight types of solid cancers, the median and interquartile range of SF2 values were, respectively, 0.56 and 0.23 [19].

Treatment of sub-confluent adherent cultures of all eight cell lines with one dose of 7.5 or 15 Gy was lethal to a large fraction of cells, which detached from the culture substratum over a period of 1–7 days after radiation. As assessed by staining for the *7*-amino-actinomycin D (7AAD) cell viability dye, 7.5 and 15 Gy radiation reduced viability among adherent cells four days later by 1%–13% and 4%–23%, respectively (Appendix A). The effect on cell viability was larger for the HNSCC compared to the NSCLC cell lines and had no association with cell doubling time or SF2 values of the cell lines.

To test whether the increase in recognition of cancer cells’ NY-ESO-1 by T cells following radiation occurs because of an increase in NY-ESO-1 protein expression on cells, we irradiated the eight cell lines with a single 15 Gy dose and after four days measured NY-ESO-1 by indirect immunofluorescence flow cytometry. While NY-ESO-1 is an intracellular protein, and the NY-ESO-1 antigenic peptides presented by MHC I are generated from it within the cell, the protein may be detectable on the cell surface [17]. Radiation led to an increased binding of the anti-NY-ESO-1 mouse IgG1 monoclonal antibody in all cell lines (Figure 2). However, a similarly increased binding was also seen for a negative control antibody (IgG1 fraction of normal mouse serum). The increased binding of both the anti-NY-ESO-1 and control antibodies was radiation-dose-dependent, with less binding at 7.5 Gy compared to 15 Gy (Appendix A). After normalization to signal with the control antibody, an increase in cell surface expression of NY-ESO-1 in irradiated cells, which was minor, was observed for only FaDu and SCC4, but not H522 or the other cell lines. We were unable to detect NY-ESO-1 in untreated or 15-Gy-irradiated H522, or A549, H460, and SCC4 cells by immunoblotting of whole cell detergent lysates (15 µg protein), which may have been because of limited sensitivity of the assay.

We also examined the effect of radiation on cell surface protein expression for two other tumor-associated proteins, calreticulin and mucin 1 (MUC1) [26,27], using direct immunofluorescence flow cytometry. Similarly to NY-ESO-1, radiation treatment of cells increased cellular binding of antibodies against both proteins in a dose-dependent manner in all eight cell lines (Appendix A). However, such dose-dependent binding was also observed for two other negative control antibodies, against the CD23 (FcεRII) IgE receptor and the CD206 (MRC1) carbohydrate-binding lectin proteins that are normally expressed only on immune cells (Appendix A shows the CD23 data). We therefore concluded that irradiated cells bind antibodies more compared to non-irradiated cells regardless of antibody specificity.

### 2.3. Irradiated Cancer Cells Were Larger Because of Mitosis Arrest and Polyploidization

A larger cell surface could be the cause of the increased non-specific binding of antibodies to irradiated cells that we observed. Indeed, light microscopy of adherent cells showed that irradiated cells of all eight cell lines were significantly larger in size compared to cells that did not receive radiation (Figure 3). Larger size was also evident in examination of forward scatter values in flow cytometry of cells in suspension (Appendix A).

Besides a larger cell size, irradiated cells also had larger nuclei and were frequently multi-nucleated, as expected with radiation-induced cell cycle arrest. These effects of radiation were also seen with a 7.5 Gy dose. DNA content analysis of some of the cell lines by fluorescence microscopy as well as cytometry confirmed that radiation treatment caused arrest at G2/M phase, and the occurrence of polyploid multinucleate cell formation because of failure of cytokinesis following karyokinesis (Figure 4).

### 2.4. Radiation Had a Greater Effect on Gene Expression of Cells That Were More Proliferative

Due to its significant effect on cell size, we could not use flow cytometry to rigorously interrogate how radiation affected the expression of tumor-associated antigens like NY-ESO-1. We therefore profiled the gene expression in cells to understand how radiation might enhance the immune recognition of such antigens. In three independent experiments, cells of seven of the eight cell lines (excluding H226) were treated with one dose of 15 Gy X-rays. Concurrent cell cultures that were not irradiated were used as controls. Gene expression in presumably live cells (adherent cells) four days after radiation treatment was examined by RNA sequencing. The duration of four days was chosen because the effect of radiation on NY-ESO-1-mediated T-cell recognition was observed with it for A498 [17], and H522 and OE19 (Figure 1) cells. We also wanted the cells to undergo at least one doubling, because of the expected effect of radiation on cell-cycle-related processes, and the largest doubling time among the seven cell lines was 61–96 h (Table 1).

Between 16,770 and 20,912 genes were identified as being expressed in each cell line. As expected, the tissue of cancer origin was reflected in the transcriptomes of untreated cell lines. In assessment of similarity of global gene expression of the cell lines with unsupervised hierarchical clustering and multidimensional scaling plots, the two groups of HNSCC and NSCLC cell lines segregated from each other (Figure 5A,B). An effect of radiation was evident in these plots for all cell lines. Expressions of 867–4121 and 734–3492 genes were significantly up- and downregulated by ≥ 1.5×, respectively, by radiation at false discovery rate (FDR) ≤ 0.05 (Figure 5C and Appendix A). A larger portion of the transcriptome (28%–37% of genes) was dysregulated by radiation in the fast-growing A549, H460, and H1299 cells with a low doubling time (Table 1), compared to the slow-growing FaDu, H522, HSC2, and SCC4 cell lines (10%–18%).

Only a small fraction of genes was affected by radiation in same manner in all seven cell lines. There were 102 downregulated genes in common (Appendix A), 44 of which encode histones and another 31 of which encode other nuclear proteins such as lamin and high mobility group proteins. Twelve of the downregulated genes encode mitochondrial proteins, including adenylate kinase 2, cytochrome C1, and fumarate hydratase. Thirty-eight genes were similarly upregulated by radiation in all cell lines (Table 2). Among their encoded proteins, 14 are nuclear, and 15 are cell surface proteins and/or secreted into extracellular space. None of the commonly upregulated genes encodes a known tumor-associated antigen. Fold-change and *p* values from differential expression analysis of each cell line are provided in Appendix A.

We validated our determinations of RNA-sequencing-based gene expression changes for four cell lines by using reverse transcription PCR to measure in the same RNA preparations the transcript levels of six genes (Figure 6A).

### 2.5. Gene Sets Downregulated by Radiation Were Predominantly Proliferation-Related

For a broader perspective on the radiation-induced transcriptomic changes, we examined gene expression at the level of gene sets. Specifically, expression for the 824 gene sets of the Molecular Signatures Database Hallmark and Reactome collections was scored using the gene set variation analysis method [28], and scores of groups of untreated and 15-Gy-irradiated cells were compared using a paired moderated t test. This gene set enrichment analysis revealed 21 sets for which enrichment was significant (nominal *p* ≤ 0.05) and similar across all cell lines (Figure 6B). Twenty of these sets had enrichment in non-irradiated cells. Nine of the 20 were cell-proliferation-related sets. Another three gene sets were involved in transcription by RNA polymerases I and III, which transcribe ribosomal DNA and transfer RNA genes that are essential for protein translation. Enrichment in irradiated cells was seen for only one set, the Reactome pathway for lipid transport mediated by high-density lipoproteins. These observations suggest that the primary transcriptomic effect of radiation was downregulation of genes involved in cell proliferation and protein translation.

### 2.6. Radiation Enhanced Expression of Subsets of Antigen Processing and Presentation Genes

No immune-related gene set was identified as significant in our gene set enrichment analysis. However, in differential gene expression analysis, radiation was noted to upregulate the expression of a number of antigen presentation genes in all cell lines—MHC I-encoding *HLA-B* and *HLA-C*, and *MR1*, which encodes an MHC I-like antigen-presenting protein. We therefore finely examined the effect of radiation on the expression of 182 genes involved in antigen processing or presentation [29]. Broadly speaking, these genes can be categorized by the molecular functions of the proteins that they encode: generation of peptide antigens from proteins through the ubiquitin-proteasome mechanism or through proteases like cathepsins; intracellular transport of antigens for loading on antigen-presenting proteins; antigen presentation; and those affecting transcription, protein folding, or trafficking within cell or to/from cell surface of other antigen processing and presentation genes.

Except for six genes, at least one of the seven cell lines expressed one of the 182 genes. While expression of only a few of the genes were similarly affected by radiation in all cell lines, examination of genes grouped by function revealed that, in general, radiation downregulated genes involved in proteasomal antigen generation or trafficking (Figure 7A). On the other hand, it upregulated genes involved in generation of antigens by standalone proteases, or transport or presentation of antigens. As previously shown in Figure 6A, the radiation-induced changes for five of these genes were validated by RT-PCR. For three cell lines, we examined cell surface MHC I protein expression by flow cytometry, and could corroborate the radiation-induced upregulation of MHC I at cell surface (Figure 7B).

## 3. Discussion

The data that we have presented here demonstrate that the NY-ESO-1 tumor-antigen-mediated T-cell immunogenicity of cancer cells is enhanced by radiation. This effect is not due to any radiation-mediated increase in gene expression for the antigen, but may be because of upregulation of its presentation consequent to an increase in expression levels of antigen processing/presentation genes, such as HLA genes encoding for class I MHC molecules that present the antigen to T cells. The upregulation of MHC genes and cell-size enlargement that we describe in our study appear to be ubiquitous sequelae of radiation across multiple cancers.

The effects of ionizing radiation on cancer cells are multifaceted [30]. They develop with different time courses, and depend on the radioresistance state of cells and the type and amount of radiation. DNA damage by ionizing radiation, the primary molecular mechanism of cellular killing by radiotherapy, occurs either directly or indirectly through intermediate reactive species, and both are equally important for X-rays because of its low linear energy transfer [31]. When DNA damage is not severe enough for immediate lethality, other effects of radiation get the chance to play out, many of which underlie the anti-tumor benefit of sublethal radiation. In the patient setting, not only cancer cells but also cells in the tumor microenvironment are irradiated. The effect on the immune component of the microenvironment is a significant contributor to the therapeutic value of radiotherapy [32]. Sublethal radiation also boosts the anti-cancer immune response by enhancing the immunogenicity of cancer cells, because of which radiotherapy is increasingly considered a powerful adjuvant for immunotherapy [33,34,35]. By examining multiple cancer cell lines, we sought to identify universal mechanisms that make irradiated cells more immunogenic.

Using cancer cell–T cell co-culture assays, we previously demonstrated that irradiated A498 renal carcinoma cells are recognized better by NY-ESO-1-specific CD8^+^ T cells [17]. In this study, we found that the same phenomenon occurs for H522 lung and OE19 esophageal cancer cells (Figure 1). It is possible that the larger cell surface of irradiated cells facilitates their recognition by T cells in co-culture assays. Nevertheless, the enhanced recognition is of significance in itself. NY-ESO-1 RNA level in H522 cells was not affected by radiation, indicating that the elevated NY-ESO-1-mediated immunogenicity of the cells was not because of a radiation-induced increase in gene expression. A broader examination of the expression levels of more than 100 genes encoding tumor antigens of cancer/testis type also showed the absence of a consistent or uniform transcription-level effect of radiation on tumor antigens across multiple cell lines. This observation is consistent with other studies [36].

Radiation has been shown to increase recognition by and subsequent activation of antigen-specific CD8^+^ T cells for H522 cell for Brachyury, carcinoembryonic antigen (CEA), and MUC1 tumor antigens. For the same three antigens, a 10 Gy dose was found to enhance CD8^+^-T-cell-mediated cytotoxicity for six breast, lung, and prostate cancer cell lines four days after irradiation [37]. Numerous other studies have also demonstrated enhanced CD8^+^-T-cell-mediated killing of sublethally irradiated cancer cells (e.g., reference [38]). Because we had access to only one tumor-antigen-specific CD8^+^ T cell clone, we could not examine whether the radiation-mediated enhancement of cancer cell recognition through MHC I-presented antigens occurred for antigens other than NY-ESO-1. The NY-ESO-1-specific T cells we used were of the HLA-A*02 subtype. Among the nine cell lines that we studied, only three (H522, OE19, and SCC4) had the same HLA subtype, which was required for our cancer cell–CD8^+^ T cell co-culture assay. Of these cell lines, radiation’s effect on SCC4 could not be examined with the co-culture assay because of the high background signal.

The number of MHC I complexes on a cell has been estimated at about 2 × 10^5^, and it is believed that together they present only about 10,000 of the billions of unique peptide antigens that exist in a cell [39]. The presentation of a tumor antigen to T cells can be heightened if a cell generates more of either the antigen or the antigen-presenting molecule. In our transcriptome analyses of seven cell lines, while radiation was found to raise the expression of many genes that encode tumor antigens, it also reduced the expression of many other genes, and the effects differed from one cell line to another. For example, the gene for NY-ESO-1, *CTAG1B*, was downregulated by radiation in H1299 cells, but in H522 it was unaffected (Appendix A). The same effects were noticeable at the protein level in flow cytometry for cell-surface-level NY-ESO-1 (Figure 2). On the other hand, upregulated expression of MHC I-encoding genes *HLA-B* and *HLA-C* was observed in all cell lines (Table 2). Expression of the other classical MHC I gene, *HLA-A*, was increased in five of the seven examined cell lines and unaffected in the remaining two. At the cell surface protein level, MHC I was increased in all examined cell lines (Figure 7B). Additionally, the *B2M* gene that makes the β2 microglobulin component of MHC I was upregulated by radiation in six cell lines and unaffected in one.

Besides MHC I, gene expression for multiple other MHC molecules was also enhanced by radiation. These included non-classical MHC I proteins like HLA-E and HLA-F, as well as class II MHC proteins such as those encoded by *HLA-DOB* and *HLA-DQB1* genes (Figure 7A). Antigen presentation by MHC II is traditionally associated with professional antigen-presenting immune cells such as B and dendritic cells. However, MHC II molecules are also present on many cancer cells, and their expression has been associated with superior clinical outcome including responsiveness to immunotherapy [40]. Thus, radiation may improve the immune response to not only the MHC I-aware CD8^+^, but also MHC II-aware CD4^+^ T cells [41]. Gene expression for the non-classical MHC I molecule MR1 was also increased by radiation in all cell lines in our study (Table 2). Unlike classical MHC I, MR1 is not polymorphic in the population and it presents small non-peptidic antigens such as metabolites [42]. A recent study suggests that such MR1-mediated presentation of the metabolome of cancer cells is utilized by specific T cells for ubiquitous cancer targeting [43].

Multiple genes of which the products participate in generation of peptide antigens, their transport into endoplasmic reticulum, or their loading on MHC proteins were also upregulated by radiation in multiple cell lines. These included those encoding various cathepsin protease (e.g., *CTSE* and *CTSS*), peptidase (e.g., *ACE* and *ERAP2*), antigen transporter (*TAP1*, *TAP2*), and tapasin-like (*TAPBPL1*) proteins. On certain other categories of genes involved in antigen processing and presentation, radiation had a negative effect. These genes were involved in proteasomal degradation of proteins, and in endocytic/vacuolar trafficking of antigens or their progenitors. However, unlike the positive effect of radiation on antigen transport and presentation genes, the negative effect was subdued and less uniform across cell lines (Figure 7A). Additionally, in contrast to the former gene categories, genes of the negatively affected categories were those that participate in other constitutive cellular functions besides antigen processing and presentation, and their dysregulation has a more modest association with tumor immunogenicity [44,45,46].

The modulation by radiation of antigen processing and presentation genes in cancer cells that we observed has been known for both cells grown in vitro and those in patient tumors (e.g., references [17,36,47,48,49]). Our study showed the universality of this effect, although we did not examine it for normal cells, nor did we examine the seven cancer cell lines in our study for the effect at timepoints other than four days after irradiation. As expected, another universal effect of radiation in our study was on genes associated with cell proliferation. Surprisingly, less than 1% of the mRNA transcriptome was dysregulated by radiation concordantly across all cell lines, suggesting that a cell’s epigenetic landscape is the dictator of its transcriptomic response to radiation. In all cell lines that we studied, irradiation caused dramatic cell enlargement and multinucleation (Figure 3 and Figure 4, and Appendix A). This appears to be a result of endoreduplication in absence of mitotic cytokinesis [50] because of dysregulation of cell proliferation genes, although we did not ascertain the possible role of radiation-induced cell-cell fusion, which has been described for some cancer cells [51]. Cell enlargement and polyploidization because of radiation is a known phenomenon, sometimes referred to as multinucleated giant cell formation [52,53,54,55]. Our study indicated that this is a ubiquitous characteristic of epithelial cancer cells exposed to moderate radiation. As highlighted by the increased non-specific antibody binding to irradiated cells in our flow cytometry data, it also calls for caution in the design and interpretation of experiments with such cells.

## 4. Materials and Methods

### 4.1. Cell Lines and Their Culture

Human A549 (Cellosaurus [56] identifier CVCL_0023) lung adenocarcinoma (AC), FaDu (CVCL_1218) head and neck squamous cell carcinoma (HNSCC), NCI-H1299 (CVCL_0060) lung SCC, NCI-H226 (CVCL_1544) lung SCC, NCI-H460 (CVCL_0459) lung AC, and NCI-H522 (CVCL_1567) lung AC cell lines were purchased from American Type Culture Collection^®^ (Manassas, VA, USA). The *NCI* prefix has been omitted from cell line names in this publication. Human HSC2 (CVCL_1218) and SCC4 (CVCL_1684) HNSCC human cell lines from Japanese Collection of Research Bioresources Cell Bank were purchased from XenoTech^®^ (Kansas City, KS, USA). Human OE19 (CVCL_1622) esophageal AC cell line from European Collection of Authenticated Cell Cultures was purchased from Sigma^®^ (St. Louis, MO, USA). Except for FaDu, the cells were directly acquired by us from the vendors, who provided short-tandem-repeat-profiling-based confirmation of cell identities. FaDu was obtained from the laboratory of Mukund Seshadri at our institution and used without re-confirmation of cell identity. Additional information about the cell lines is provided in Table 1. HLA types of the cells were noted from the TRON Cell Line Portal [57]. Cells were cultured adherent in RPMI-1640 or DMEM (high glucose) medium (Mediatech^®^, Manassas, VA, USA) with 10% *v*/*v* fetal bovine serum (FBS; VWR^®^, Radnor, PA) at 37 °C under 5% CO_2_ and 90% humidity. RPMI-1640 was used for H226, H522, H1299, and OE19. For OE19, the medium had 2 mM additional glutamine. Cell culture plates were rinsed with phosphate-buffered saline (PBS; 137 mM NaCl, 2.7 mM KCl, 10 mM Na_2_HPO_4_ and 1.8 mM KH_2_PO4, pH 7.4) prior to cell harvesting so that only adherent cells were collected. Trypsin (0.05% *w*/*v*) with 0.25 mM ethylene-diamine-tetra-acetic acid (EDTA) (Gibco^®^, Grand Island, NY, USA) was used for maintaining cell cultures. All experiments with cells were performed within 10 weeks of total culture time since acquisition.

### 4.2. Radiation Treatment of Cells

Cells were plated in 10 cm polystyrene plates for tissue culture in 8 mL of medium for radiation treatment. Culture medium was refreshed if the medium on plates was more than two days old. Plates with cells that did not receive radiation were concurrently set up. Cells were at 65%–85% confluence at the time of irradiation, which was performed by placing the plates inside a calibrated RX-150 cabinet system (Faxitron^®^, Lincolnshire, IL, USA) with a self-rectifying, thermionic X-ray tube (130 kVp max) equipped with a beryllium window. Radiation treatment was done at room temperature and at a dose rate of 0.54 Gy/min. Irradiated plates were immediately returned to cell culture incubator. After a day, cells of both control and irradiated plates were detached and re-plated so that they were sub-confluent at time of harvest two or three days later. Cell culture medium was refreshed a day before harvest.

### 4.3. NY-ESO-1-Specific T-Cell Activation Assay

Adherent cancer cells were detached from tissue culture plates using trypsin-EDTA and washed with PBS. Cells (2.5 × 10^4^) were then co-cultured with 5 × 10^3^ human, HLA-A*02-restricted, NY-ESO-1-specific human CD8^+^ T cells [15] in 0.2 mL of serum-free RPMI-1640 medium per well of 96 well plates (product MAHAS4510, Millipore^®^, Billerica, MA, USA) that had been coated with an antibody against IFNγ (details below). After 24 h of culture, the plates were processed for ELIspot assay with streptavidin-conjugated alkaline phosphatase (product 3310-10, Mabtech^®^, Nacka Strand, Sweden) and 5-bromo-4-chloro-3-indolyl-phosphate/nitro blue tetrazolium (Sigma^®^) to count the number of IFNγ-secreting cells on the plates using a CTL ImmunoSpot™ Analyzer (Cellular Technologies^®^, Cleveland, OH, USA). Average counts from triplicate wells were used for analyses. The maximum count for wells with only T cells was 1 in all experiments.

### 4.4. Antibodies

Primary antibodies used in this study and their human targets were as follows. Calnexin: Alexa Fluor™ 488 conjugate of mouse IgG2a mAb clone 4F10 (product M178-A48, MBL^®^, Woburn, MA, USA), used at 1:150 dilution; calreticulin: phycoerythrin (PE)-conjugate of mouse IgG1 mAb clone FMC 75 (product ADI-SPA-601PE-D, Enzo^®^, Farmingdale, NY, USA), used at 1:100 dilution; CD206: PE conjugate of mouse IgG1 mAb clone DCN228 (product number 130-095-220, Miltenyi Biotec^®^, Bergisch Gladbach, Germany), used at 1:100 dilution; CD23: fluorescein isothiocyanate (FITC) conjugate of mouse IgG1 mAb clone EBVCS2 (product 11-0238-42, Invitrogen^®^, Carlsbad, CA, USA), used at 1:100 dilution; IFNγ: mouse IgG1 mAb clone 1-D1K (product 3420-3-1000, Mabtech^®^), used at 15 µg/mL, and biotinymated mouse IgG1 mAb clone 7-B6-1 (product 3420-6-250, Mabtech^®^), used at 1 µg/mL; MHC I: PE conjugate of mouse IgG2a mAb clone Tu149 (product MHBC04, Invitrogen^®^), used at 1:100 dilution; MUC1: FITC conjugate of mouse IgG3 mAb (product LS-B2244, LifeSpan BioSciences^®^, Seattle, WA, USA), used at 1:100 dilution; NY-ESO-1: mouse IgG1 mAb clone E978 (product 356200, Invitrogen^®^) used at 1:250 dilution. PE-conjugated rat IgG1 mAb clone A85-1 against mouse IgG1 (product 550083) was from BD Biosciences^®^ (San Jose, CA, USA) and used at 1:100 dilution. AffiniPure™ Fcγ fragment of goat anti-human IgG antibody (product 109-005-098) was from Jackson ImmunoResearch^®^ (West Grove, PA, USA) and used at 2 µg/mL.

### 4.5. Flow Cytometry

Adherent cells were detached from tissue culture plates using trypsin-EDTA, washed with PBS, and resuspended at 1 × 10^6^ cells/mL in Hank’s balanced salt solution (HBSS; Mediatech^®^) with 0.5% w/v bovine serum albumin (BSA; Amresco^®^, Solon, OH, USA) and 2 µg/mL Fcγ fragment of goat anti-human IgG antibody. After incubation at 4 °C for 30 min, a primary antibody (details below) was added to the cell suspension (100 µL) which was then incubated for another 30 min at 4 °C. If the antibody did not have a fluorophore, cells were spun down at 450 g for 2 min, resuspended to 100 µl HBSS with 0.5% w/v BSA, and incubated with a fluorophore-conjugated secondary antibody (details below) at 4 °C for 30 min. Cells were finally spun down at 450× *g* for 2 min and resuspended in 200 µL HBSS with 0.5% *w*/*v* BSA for flow cytometry on a FACSCalibur™ machine (BD Biosciences^®^). Intensities for forward and side scatter, and fluorescence in the FL1, FL2, and FL4 channels were recorded for 10–20 thousand events with CellQuest™ Pro software (version 5; BD Biosciences^®^). The *7*-amino-actinomycin D viability dye (Sigma^®^) was added to the cell suspension at 1 µg/mL immediately before cytometry. Cytometry data were analyzed with FCS Express™ 4 (De Novo Software^®^, Pasadena, CA, USA).

### 4.6. DNA Content Analysis

Adherent cells were detached from tissue culture plates using trypsin-EDTA, washed with PBS, and re-suspended in 70% *v*/*v* ethanol at 1–2 × 10^6^ cells/mL. After 1–4 days at 4 °C, cells were washed twice with PBS to remove ethanol, resuspended at 2 × 10^6^ cells/mL in FxCycle™ propidium iodide/RNAse Staining Solution (product F10797, Molecular Probes^®^, Eugene, OR, USA), and incubated in the dark for 30 min at room temperature before fluorescence cytometry with excitation/detection in FL2 channel of a FACSCalibur™ instrument.

### 4.7. Immunofluorescence Microscopy of Cells

Cells were grown adherent to 50%–80% confluence on eight-chambered polymer slides (product 80826, Ibidi^®^, Martinsried, Germany), whereupon they were fixed for 20 min with freshly prepared 4% *w*/*v* paraformaldehyde in PBS. Fixation and the following steps were performed at room temperature. After quenching with 2 M glycine in PBS for 15 min followed by permeabilization with 0.2% *v*/*v* Triton X-100 (Sigma^®^) for 10 min, the slides were rinsed twice with PBS. Slides were then incubated for 1 h with PBS with 5% *v*/*v* sheep serum, 0.15% *v*/*v* Triton X-100, and 1% v/v Fcγ fragment of goat anti-human IgG antibody. After this blocking step, slides were incubated for 1 h with appropriate fluorophore-conjugated primary antibody (1:150 dilution) in PBS with 5% *v*/*v* sheep serum and 0.12% *v*/*v* Triton X-100, and then washed twice with PBS with incubations of 5 min for each wash. Following a 5 min incubation with PBS with 1 µg/mL 4’,6-diamidino-2-phenylindole (DAPI; Sigma^®^), slides were washed four times and imaged using an Eclipse™ Ti microscope (Nikon^®^, Melville, NY, USA) equipped with an RT KE camera (Spot Imaging^®^, Sterling Heights, MI, USA) after adding a mounting medium (Ibidi^®^).

### 4.8. Total RNA Isolation

An affinity spin-column-based total RNA purification kit with a DNAse treatment step was used (product 17200, Norgen Biotek^®^, Thorold, ON, Canada). RNA preparations were quantified by absorbance spectrophotometry at 260 nm and with TapeStation™ RNA ScreenTape (Agilent^®^, Santa Clara, CA, USA). RNA integrity number values were in the 6.6–9.8 range (mean = 8.8; standard deviation (SD) = 0.8).

### 4.9. Reverse Transcription and PCR

Total RNA (1–5 µg) was reverse transcribed at 42 °C for 1 h in reactions of 15–20 µL with 0.5 mM dNTP mix (Promega^®^, Madison, WI, USA), 60 µM random DNA hexamers (Applied Biosystems^®^, Foster City, CA, USA), 200 U M-MuLV reverse transcriptase (New England BioLabs^®^, Ipswich, PA, USA), and 8 U murine RNase inhibitor (New England BioLabs^®^). The reactions (2%–8% volume) were directly used as template for 15 µL quantitative PCR reactions that were prepared with FastStart™ Universal SYBR Green Master (Rox) master-mix (Roche^®^, Indianapolis, IN, USA) and performed on a LightCycler™ 480 system (Roche^®^, Indianapolis, IN, USA). Quantification cycle (C_q_) values were calculated by the system’s software using the maximum second derivative method, and the mean C_q_ value of duplicate or triplicate PCR reactions was used for further analysis after subtracting the mean C_q_ value for the housekeeping gene *ACTB*, which encodes β actin. Electrophoresis of PCR reactions on agarose gel was used to confirm the generation of a single product. Forward and reverse PCR primers, designed to span introns and synthesized by Applied Biosystems^®^, were, respectively, as follows. *ACTB* (amplicon of 174 bp): AGCCTCGCCTTTGCCGA and CTGGTGCCTGGGGCG; *CALR* (102 bp): ACAACCCCGAGTATTCTCCC and TGTCAAAGATGGTGCCAGAC; *CTSS* (232 bp): GCCTGATTCTGTGGACTGG and GATGTACTGGAAAGCCGTTGT; *GAPDH* (90 bp): CATCAATGGAAATCCCATCA and GACTCCACGACGTACTCAGC; *HLA-B* (255 bp): CTACCCTGCGGAGATCA and ACAGCCAGGCCAGCAACA; *HLA-C* (98 bp): CACACCTCTCCTTTGTGACTTCAA and CCACCTCCTCACATTATGCTAACA; *TAP1* (499 bp): AGGGCTGGCTGGCTGCTTTGA and ACGTGGCCCATGGTGTTGTTAT. All primer sequences were obtained from published studies [58,59,60,61].

### 4.10. Total RNA Sequencing

Sequencing libraries were prepared from 1 µg of each total RNA sample using reagents and protocols provided in TruSeq™ Stranded Total RNA Library Prep Gold kit (Illumina^®^, San Diego, CA, USA). Ribosomal RNA depletion and 10 PCR cycles were employed during library preparation. TapeStation™ D1000 ScreenTape (Agilent^®^) and Library Quantification Kit (Kapa Biosystems^®^, Wilmington, MA, USA) assays were used to confirm good quality of libraries. All libraries for A549, H460, and H1299 were sequenced together on an Illumina^®^ HiSeq™ 2500 instrument in a two-lane, rapid-run flow cell using HiSeq™ Rapid Cluster Kit v2 - Paired-End and Rapid SBS Kit v2 reagents to obtain paired reads of 101 bases. The other libraries were sequenced together on an Illumina^®^ NextSeq™ 500 instrument in a four-lane flow cell using NextSeq™ 500/550 High Output Kit v2 reagents to obtain paired reads of 75 bases. Casava v1.8.2 software was used to demultiplex sequencing data. For each library, 17–40 million read pairs (mean = 27; SD = 6) were obtained. Raw sequencing data were deposited in European Nucleotide Archive repository with identifiers PRJEB18075 and PRJEB20193.

### 4.11. Processing of RNA Sequencing Data

Raw sequencing data were filtered with Trimmomatic v0.35 [62] to remove adapter and poor-quality sequence segments. The following software options were used in the order shown: headcrop = 12, illuminaclip = 2:30:10:6:true, leading = 5, trailing = 5, slidingwindow = 4:15, minlen = 30. The remaining read pairs (14–39 million (mean = 27; SD = 6)) were mapped against the GRCh38 reference genome and transcriptome using HISAT2 [63] (020516 release; software option k = 2) and reference files provided with it. The average overall read mapping rate was 83% (range = 61–91; SD = 6). Among mapped reads, the fraction mapping to known genes was 0.26–0.81 (mean = 0.64; SD = 0.12). Uniquely mapped reads and Ensembl gene annotations (release 81) were used to generate gene-level mapped read counts with Subread featureCounts [64] (v1.5.0-p1; using software options O and primary).

### 4.12. Analyses of Gene Expression

Differential gene expression analysis comparing irradiated and non-irradiated cells (*n* = 3 each) in paired design was performed independently for each cell line using generalized linear modeling and likelihood ratio test in edgeR [65] R Bioconductor software (v3.16.1). In all analyses, genes with low expression were excluded (read count < 8 in each of > 3 of samples). Rate of false discovery rate (FDR) arising from multiple testing was estimated using *p* values with the Benjamini–Hochberg method. Gene ontologies were examined with DAVID tools (v6.8) [66]. To determine the enrichment of expression of the 50 and 674 gene sets of the Molecular Signatures Database (mSigDB; v6.2) Hallmark and C2:CP Reactome collections, respectively, the gene set variation analysis (GSVA) method implemented in the GSVA Bioconductor package (v1.30.0) was utilized [28]. Two-group comparison of GSVA scores in paired design, performed separately for each cell line, was done with the empirical Bayes-moderated t test in limma (v3.38.2) Bioconductor package [67]. For visualization as well as gene expression enrichment and correlation analyses, log_2_-transformed (with padding of 0.05) normalized gene expression values in count per million unit were used. Normalization was performed across all samples, excluding genes with read count < 8 in each of > 3 of samples, by edgeR’s trimmed mean of M-values (TMM) method. A set of 107 genes encoding cancer/testis antigens was obtained in autumn of 2017 from entries in the Cancer Testis Database (CTdatabase) [68] as the set of genes with unambiguously assignable Human Genome Organization Gene Nomenclature Committee (HGNC) gene symbols. A set of 182 genes related to antigen processing and presentation was collated from the mSigDb C5:BP GO_ANTIGEN_PROCESSING_AND_PRESENTATION gene set (v7.0) and published literature [29].

### 4.13. Other

Cells in suspensions were counted with a Scepter™ electronic counter with 60 µm sensor (Millipore^®^). Light microscopy of cells was performed on an Axio™ Observer microscope (Zeiss^®^, Maple Grove, MN, USA) with an EOS 450D digital camera (Canon^®^, Lake Success, NY, USA). Unless noted otherwise, statistical tests were two-tailed and threshold of 0.05 was used to deem significance of their *p* values.

## Figures and Tables

**Figure 1 ijms-21-02573-f001:**
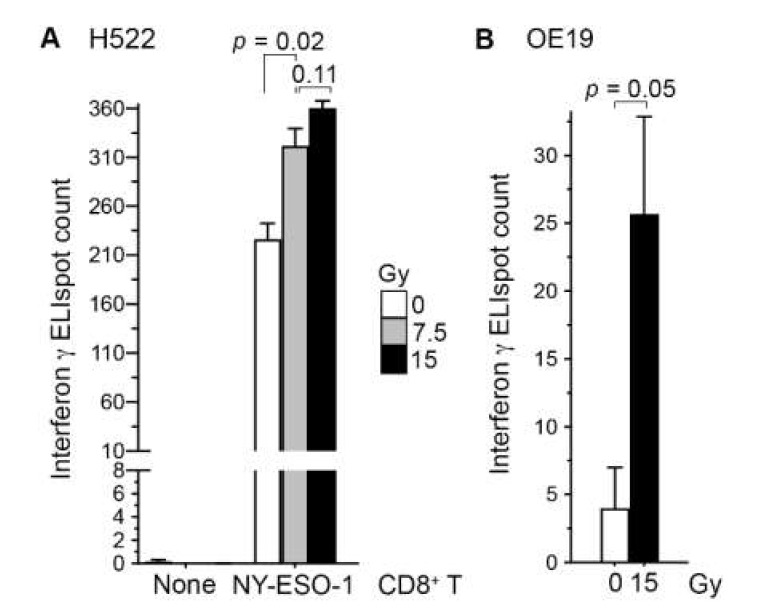
Irradiation of cancer cells enhanced their recognition by antigen-specific CD8^+^ T cells. Human H522 lung (**A**) or OE19 esophageal (**B**) adenocarcinoma cells were irradiated with one dose of 7.5 or 15 Gy X-rays or left untreated (0 Gy). Three days later, adherent cells were collected and co-cultured in triplicate at a 5:1 ratio with or without NY-ESO-1-specific human CD8^+^ T cells on an ELISpot plate for detecting interferon-γ-producing cells after a day. The mean and its standard error are plotted, and *p* values in standard t tests are shown.

**Figure 2 ijms-21-02573-f002:**
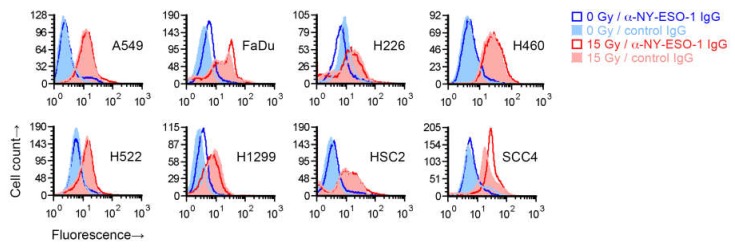
Effect of radiation on cell surface NY-ESO-1 measurement by fluorescence flow cytometry. Sub-confluent adherent cultures of indicated cell lines that had been grown in parallel to same cell density were treated with 0 or 15 Gy X-rays. After four days, adherent cells were collected by scraping and examined by flow cytometry for binding of an unconjugated mouse IgG1 antibody against NY-ESO-1. Antibody binding was indirectly detected with a fluorophore-conjugated rat antibody against mouse IgG. Binding of a control IgG1 (normal mouse serum fraction) was similarly detected. Shown are representative histograms of viable cells identified by *7*-amino-actinomycin D staining.

**Figure 3 ijms-21-02573-f003:**
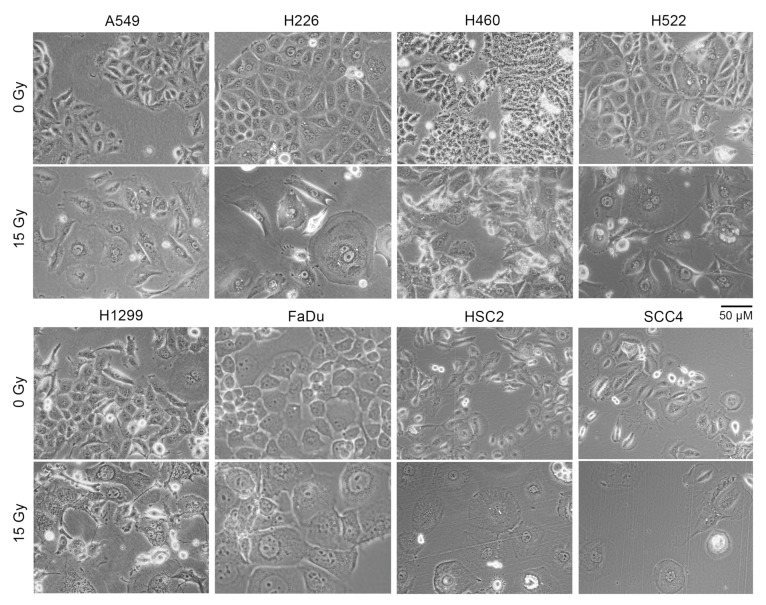
Cell size increased following radiation treatment. Indicated human cancer cell lines were treated with one dose of 15 Gy X-rays or left untreated (0 Gy). Shown are representative phase contrast light microscopy images of cells after four days. Floating cells were removed before imaging.

**Figure 4 ijms-21-02573-f004:**
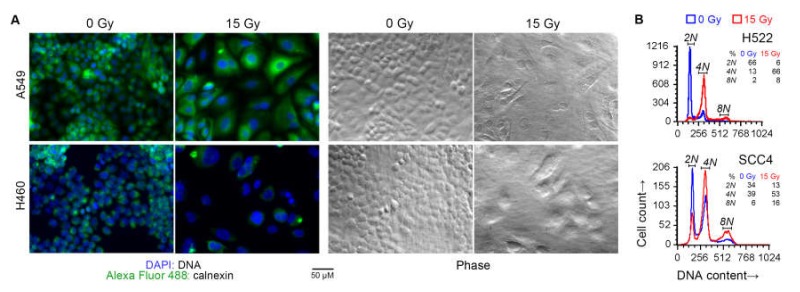
Cell cycle arrest following radiation treatment. Indicated human cancer cell lines were treated with one dose of 15 Gy X-rays or left untreated (0 Gy). After four days, the DNA content of adherent cells was examined. (**A**) Overlaid green and blue fluorescence and phase contrast light microscopy images of representative fields are shown. Cells were fixed and permeabilized, and stained with an Alexa Fluor 488 green-fluorophore-conjugated monoclonal antibody against the endoplasmic-reticulum-resident calnexin protein, and the blue fluorescent DNA-binding dye 4’,6-diamidino-*2*-phenylindole (DAPI). (**B**) Histograms of DNA content of singlet cells are shown. Adherent cells were collected by scraping, fixed, stained with propidium iodide (PI) in a buffer with RNAse, and examined by fluorescence flow cytometry. DNA content was measured as orange fluorescence from PI–DNA binding. Noted are fractions of cells with 2N, 4N, and 8N DNA content (approximately diploid, tetraploid, and octaploid).

**Figure 5 ijms-21-02573-f005:**
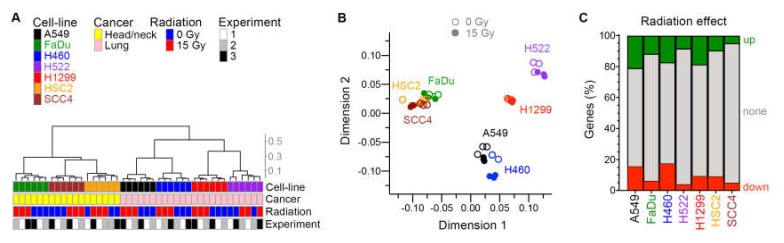
Effect of radiation on transcriptome. For seven human cancer cell lines, gene expression in adherent cells four days after radiation treatment was examined in three experiments with paired cell cultures that were or were not treated with one dose of 15 Gy X-rays. Similarity among the total 42 cellular transcriptomes is illustrated with unsupervised hierarchical clustering (**A**) and multi-dimensional scaling (**B**) plots. Cosine distance and Ward agglomeration methods were used. Inter-cluster distances in the dendrogram are indicated with a scale. (**C**) Effect of radiation on expression of genes in each cell line is depicted with stacked barplots that show the fractions of genes of which expression was up- or down-regulated ≥1.5-fold at 0.05 false discovery rate in paired likelihood ratio test in edgeR.

**Figure 6 ijms-21-02573-f006:**
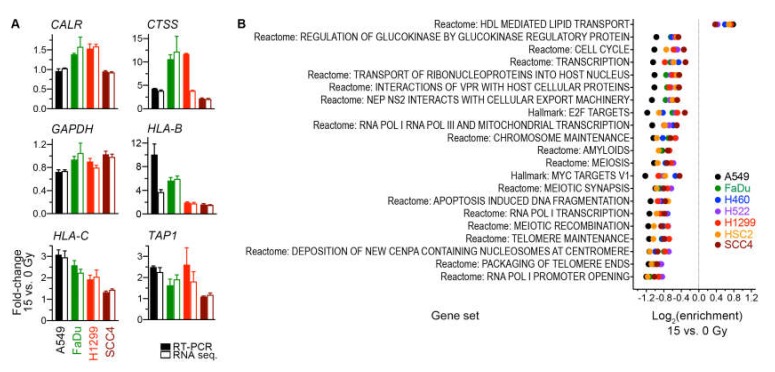
(**A**) Validation by reverse transcription (RT)-PCR of radiation-induced gene expression changes that were determined from RNA sequencing data. Mean of fold-change values and its standard error for pairs of 15-Gy-treated and untreated cells of three independent experiments are shown for six genes. The same RNA preparations were used for both RNA sequencing and RT-PCR. Global gene expression measurements by RNA sequencing were processed with the trimmed median of M-values method into count per million values. All values were normalized against those for the housekeeping *ACTB* gene prior to fold-change calculations. (**B**) Analyses of radiation-induced gene expression changes at the gene set level. Enrichment for 21 significant gene sets (*p* < 0.05) with similar enrichment across all seven indicated cell lines is shown. Gene expression of the 15-Gy-treated and untreated cells for Molecular Signature Database Hallmark and Reactome gene sets was scored using the gene set variation analysis method. Enrichment for a gene set was calculated as the ratio of scores of the groups of cells, and its statistical significance was estimated with a paired, empirical Bayes-moderated t test.

**Figure 7 ijms-21-02573-f007:**
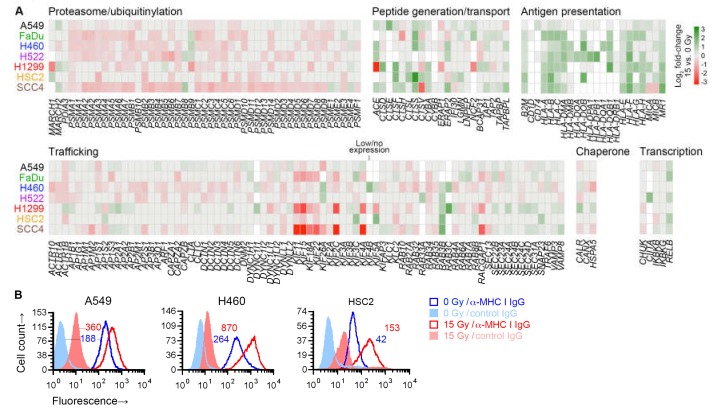
Effect of radiation on antigen processing and presentation genes. (**A**) Heatmap shows effect of 15 Gy radiation (log_2_ fold-change, compared to untreated cells (0 Gy)) on expression in indicated cell lines of 176 genes involved in antigen processing and presentation, of which expression was detected in at least one cell line. Genes are grouped by their function. Genes for which there was no effect (nominal *p* ≥ 0.05 in paired likelihood ratio test in edgeR) are shown in grey. Those with *p* < 0.05 are colored as per the displayed scale. Genes of which the expression was considered too low were not examined and are shown in white. (**B**) Cell surface expression of class I major histocompatibility complex (MHC I) proteins on viable 15-Gy-treated and untreated cells is depicted for three cell lines with representative direct fluorescence flow cytometry histograms. Viable cells were identified by *7*-amino-actinomycin D staining. MHC I proteins were detected with a phycoerythrin (PE)-conjugated monoclonal IgG antibody. Separate portions of the same cell samples were examined for binding of a PE-conjugated negative control IgG. Geometric mean fluorescence intensity values for MHC I normalized against the negative control by subtraction are noted. For both *A* and *B*, sub-confluent adherent cultures of indicated cell lines that had been grown in parallel to same cell density were treated with one dose of 0 or 15 Gy X-rays. After four days, adherent cells were collected by scraping and examined.

**Table 1 ijms-21-02573-t001:** Characteristics of the head and neck, and lung cancer cell lines used in this study ^a,b^.

	Parent Tissue	Parent Cancer	Patient Demographic	Karyotype Modal Number	Doubling Time ^c^ (Hours)	SF2 ^d^	SF8
A549	Lung	Lung AC	White M, 58 y	66	22	0.68	0.05
FaDu	Hypopharynx	Hypopharynx SCC	White M, 56 y	64	50	0.72	
H226	Metastatic pleural effusion	Lung SCC	M	47	64	0.63	0.12
H460	Metastatic pleural effusion	Lung LCC	M	57	23	0.75	0.25
H522	Lung	Lung AC	White M, 58 y	53	50	0.43	0.003
H1299	Metastatic lymph node	Lung NSCLC	White M, 43 y		30	0.52	
HSC2	Mouth	Mouth SCC	M, 69 y		27–38	0.66	
SCC4	Tongue	Tongue SCC	M, 55 y	50–80	61–96	0.59	

^a^AC: adenocarcinoma; LCC: large cell carcinoma; M: male; NSCLC: non-small-cell lung cancer; SCC: squamous cell carcinoma; SF2 and SF8: clonogenic radiation survival fraction at 2 and 8 Gy respectively; y: age in years. ^b^ Information from web pages of Japanese Collection of Research Bioresources Cell Bank (HSC2 and SCC4), and American Type Culture Collection^®^ (others) unless noted otherwise. ^c^ From Rangan et al. for Fadu [20], Cowley et al. for H1299 [21], and Amundson et al. for H226, H460, and H522 [19]. Doubling times observed for HSC2 and SCC4 in this study were about 1.5 and 3 days, respectively. ^d^ SF2 and SF8 values from Shao et al. for FaDu [22], Song et al. for H1299 [23], Shintani et al. for HSC2 [24], Zhang et al. for SCC4 [25], and Amundson et al. for others [19].

**Table 2 ijms-21-02573-t002:** Genes for which expression was upregulated by radiation in all cell lines^a^.

Gene	Description	Log_2_ Fold-Change	P
*ABTB2*	Ankyrin repeat and BTB (POZ) domain containing 2	0.3–1.9	2.3 × 10^−43^–1.4 × 10^−2^
*AMPD3*	Adenosine monophosphate deaminase 3	0.6–1.6	2.0 × 10^−30^–2.6 × 10^−3^
*BCL6*	B-cell CLL/lymphoma 6	0.3–1.9	1.2 × 10^−5^–1.2 × 10^−2^
*BTBD19*	BTB (POZ) domain containing 19	0.6–2.4	1.2 × 10^−29^–1.1 × 10^−2^
*CDKN1A*	Cyclin-dependent kinase inhibitor 1A (p21, Cip1)	0.9–3.0	2.1 × 10^−165^–1.7 × 10^−5^
*COL1A1*	Collagen, type I, alpha 1	0.6–1.9	1.3 × 10^−13^–1.2 × 10^−3^
*COL7A1*	Collagen, type VII, alpha 1	0.4–2.5	1.5 × 10^−35^–3.0 × 10^−2^
*DYSF*	Dysferlin	0.2–3.1	1.7 × 10^−25^–4.1 × 10^−2^
*FAM214A*	Family with sequence similarity 214, member A	0.3–1.8	3.9 × 10^−50^–2.1 × 10^−2^
*FAM71F2*	Family with sequence similarity 71, member F2	0.4–2.5	4.8 × 10^−21^–3.2 × 10^−2^
*FBXO32*	F-box protein 32	0.6–2.2	4.6 × 10^−43^–4.6 × 10^−2^
*HAP1*	Huntingtin-associated protein 1	0.7–1.8	1.4 × 10^−8^–1.1 × 10^−2^
*HIVEP1*	Human immunodeficiency virus type I enhancer binding protein 1	0.4–0.9	6.3 × 10^−9^–1.6 × 10^−2^
*HIVEP2*	Human immunodeficiency virus type I enhancer binding protein 2	0.7–1.4	4.5 × 10^−25^–8.0 × 10^−9^
*HLA-B*	Major histocompatibility complex, class I, B	0.4–1.9	1.3 × 10^−33^–4.5 × 10^−6^
*HLA-C*	Major histocompatibility complex, class I, C	0.3–1.4	2.7 × 10^−22^–2.2 × 10^−2^
*LAMA3*	Laminin, alpha 3	0.4–1.3	8.7 × 10^−13^–8.4 × 10^−3^
*LIPH*	Lipase, member H	0.4–2.2	7.9 × 10^−12^–7.5 × 10^−3^
*LSMEM1*	Leucine-rich single-pass membrane protein 1	0.8–1.7	7.6 × 10^−10^–4.6 × 10^−2^
*LTBP3*	Latent transforming growth factor beta binding protein 3	0.4–1.4	5.2 × 10^−60^–8.3 × 10^−3^
*MCAM*	Melanoma cell adhesion molecule	0.7–3.6	2.1 × 10^−25^–9.9 × 10^−3^
*MR1*	Major histocompatibility complex, class I-related	0.2–2.0	3.7 × 10^−93^–3.3 × 10^−2^
*NEB*	Nebulin	0.3–2.8	7.6 × 10^−24^–3.4 × 10^−2^
*NEK10*	NIMA-related kinase 10	0.9–1.9	3.6 × 10^−35^–3.7 × 10^−3^
*NOTCH2NL*	Notch 2 N-terminal like	0.4–1.1	1.1 × 10^−9^–4.0 × 10^−2^
*PLA2G4C*	Phospholipase A2, group IVC (cytosolic, calcium-independent)	0.8–2.7	5.0 × 10^−30^–2.1 × 10^−3^
*PLEKHA6*	Pleckstrin homology domain containing, family A member 6	0.6–3.8	2.1 × 10^−18^–1.2 × 10^−2^
*PRDM1*	PR domain containing 1, with ZNF domain	0.5–2.4	3.2 × 10^−25^–2.6 × 10^−3^
*RGL1*	Ral guanine nucleotide dissociation stimulator-like 1	0.4–2.4	1.9 × 10^−98^–4.3 × 10^−2^
*RP11-288H12.3*	Calcineurin like EF-hand protein 1 pseudogene (LOC729603)	0.8–1.1	2.2 × 10^−5^–1.7 × 10^−2^
*RP11-734K2.4*	POC1B antisense RNA 1 (POC1B-AS1)	0.8–1.8	1.4 × 10^−9^–8.4 × 10^−4^
*SERPINE1*	Serpin peptidase inhibitor, clade E, member 1	0.7–2.0	8.6 × 10^−19^–2.2 × 10^−4^
*SERPINE2*	Serpin peptidase inhibitor, clade E, member 2	0.7–2.8	8.1 × 10^−58^–4.2 × 10^−4^
*SGK1*	Serum/glucocorticoid regulated kinase 1	0.5–3.2	2.8 × 10^−25^–8.8 × 10^−3^
*STX3*	Syntaxin 3	0.4–1.6	4.7 × 10^−15^–1.2 × 10^−2^
*TAGLN*	Transgelin	0.2–3.2	5.3 × 10^−37^–1.8 × 10^−2^
*TNFAIP3*	Tumor necrosis factor, alpha-induced protein 3	1.0–2.7	1.1 × 10^−28^–1.5 × 10^−3^
*TRIB1*	Tribbles pseudokinase 1	0.4–0.9	4.6 × 10^−10^–2.9 × 10^−2^

^a^Gene expression differences between three pairs of 15-Gy-irradiated and non-irradiated cells of seven cell lines was analyzed via paired likelihood ratio test in edgeR software. Genes upregulated by radiation with nominal *p* ≤ 0.05 in all cell lines are listed along with ranges of log_2_ fold-change (15 Gy vs. 0 Gy) and *p* values among the cell lines.

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
