# Peer review of "Sublethal Radiation Affects Antigen Processing and Presentation Genes to Enhance Immunogenicity of Cancer Cells"

_ijms, 2020, doi:10.3390/ijms21072573_

Round 1
Reviewer 1 Report
In their manuscript „Sublethal radiation affects antigen processing and presentation genes to enhance immunogenicity of cancer cells“, the authors present the interesting observation that sublethal radiation boosts the immune presentation machinery leading to increased recognition by tumor antigen specific T cells.
This straightforward study presents a new mechanism to explain the immunogenic properties of radiotherapy and the positive effects on immunotherapy. However, it would have been interesting to know whether this is the result of a general reprogramming of cellular physiology by radiation also in normal cells or restricted to cancer cells.
Author Response
Manuscript ijms-756185: Sublethal radiation affects antigen processing and presentation genes to enhance immunogenicity of cancer cells
Response to Reviewer 1
Reviewer: In their manuscript „Sublethal radiation affects antigen processing and presentation genes to enhance immunogenicity of cancer cells“, the authors present the interesting observation that sublethal radiation boosts the immune presentation machinery leading to increased recognition by tumor antigen specific T cells.
This straightforward study presents a new mechanism to explain the immunogenic properties of radiotherapy and the positive effects on immunotherapy. However, it would have been interesting to know whether this is the result of a general reprogramming of cellular physiology by radiation also in normal cells or restricted to cancer cells.
Response: We thank the reviewer for taking a critical look at our manuscript. We agree that it will be interesting to know if the radiation-mediated immunogenicity enhancement that we observed for cancer cells also occurs in normal cells. We have now added this point in Discussion.
Our study did not include normal cells. We also could not find research literature or publicly available data to potentially answer this question for normal human epithelial cells examined at radiation dose and time-course similar to those of our study.
Reviewer 2 Report
The purpose of the manuscript presented by Punnanitinont et al. was the assessment of the impact of ionizing radiation on antigen presentation in a variety of human cancer lines. More particularly, authors report on increased New York esophageal squamous cell carcinoma 1 (NY-29 ESO-1) tumor antigen recognition by NY-ESO-1-specific CD8+ T cells in esophagus and lung cells following a 7.5 and 15 Gy exposure in line with an enlargement and multinucleation/polyploidy of the cells at four days after irradiation. In addition, they indicate an up regulation of genes of antigen processing and presentation pathways in all cell lines.
Basic mechanism(s) implicated in the enhancement of antigen presentation by ionizing radiation in malignant cells cover an interesting, clinically relevant and up-to-date focus of research. The manuscript, however, suffers from a multitude of substantial shortcomings and inaccuracies as listed successive that highly weaken the enthusiasm for the investigation.
Major points of criticism:
- Figure 1 is restricted to two lines with a huge difference in IFN-gamma detection most pronounced in H522 cells. By contrast, authors indicate a lack of NY-ESO-1 surface detection in H255 cells (page 4, line 122-123) that is contradictory for activation of antigen specific T cells and has to be discussed.
- In line with that, a major concern with the results is the notation that negative control antibody display comparable binding and kinetics after irradiation and authors stated not to be able to detect 122 NY-ESO-1 in 15 Gy irradiated H522, A549, H460, and SCC4 cells by immunoblotting. Authors should include positive controls, e.g. radiation induced up regulation of checkpoint PD-1 expression on tumor cell surface.
- Analyses presented were performed at four days after radiation exposure which in principle is a long period of time. Accordingly, rationales for choosing that late time point should be given and discussed in detail.
- Authors claimed to analyze mechanisms of antigen presentation, but some figures (Fig. 3, Fig. 4) mainly include assessment of proliferation and morphological characterization and describe well known phenomenons like cell cycle arrest and polyploidy in all the cell lines used. These data neither contribute novel findings nor did they include data important for the major purpose of the manuscript. Thus, these data should be removed.
- Table 1 mainly covers published data on well-established cell lines derived from lung and head and neck malignancies but does not contain findings from the present study. Thus, the table should be moved from the results section to a supplemental file.
- Table 2. Gene expression differences between three pairs of 15 Gy irradiated and non-irradiated cells of seven cell lines were depicted. Authors clearly should give selection criteria for these lines and reasons not to include all lines in comparisons.
- Figure 7B. Authors failed to display MHC expression on H522 cells to refer to the data given in figure 1, hampering to follow their statement (page 11, line 4): “… increase in gene expression for the antigen but likely because of up-regulation by radiation of HLA genes encoding for class I MHC molecules that present the antigen to T cells”.
Author Response
Manuscript ijms-756185: Sublethal radiation affects antigen processing and presentation genes to enhance immunogenicity of cancer cells
Response to Reviewer 2
Reviewer: Figure 1 is restricted to two lines with a huge difference in IFN-gamma detection most pronounced in H522 cells. By contrast, authors indicate a lack of NY-ESO-1 surface detection in H255 cells (page 4, line 122-123) that is contradictory for activation of antigen specific T cells and has to be discussed. In line with that, a major concern with the results is the notation that negative control antibody display comparable binding and kinetics after irradiation and authors stated not to be able to detect 122 NY-ESO-1 in 15 Gy irradiated H522, A549, H460, and SCC4 cells by immunoblotting. Authors should include positive controls, e.g. radiation induced up regulation of checkpoint PD-1 expression on tumor cell surface.
Response: In the cancer cell-T cell interaction, the NY-ESO-1 antigen of 8-10 amino acids together with the MHC I protein complex that ‘holds’ the antigen on the cancer cell surface is recognized by CD8 and T cell receptor proteins on the T cell surface. NY-ESO-1-mediated CD8+ T cell recognition as measured by interferon g production was significantly higher for irradiated H522 compared to irradiated OE19 cells. This does not imply that the two cells differ for their surface MHC I-presented NY-ESO-1 levels because other co-factors on or within the cancer or T cells can affect recognition of MHC I-NY-ESO-1 by the T cells and/or subsequent activation leading to interferon g production. As we note in the manuscript (reference 18), NY-ESO-1 RNA level of H522 and OE19 are similar.
Unlike for the T cell assay, the cell surface NY-ESO-1 protein measurement that we performed with flow cytometry was dependent on binding of the NY-ESO-1-specific IgG antibody to NY-ESO-1 protein on cell surface (and not necessarily the NY-ESO-1 antigen peptide within MHC I complexes which can arise from intracellular NY-ESO-1 protein). We now note this in the relevant Results section (2.2).
The flow cytometry data in Figure 2 shows that untreated as well as irradiated H522 cells bound the a-NY-ESO-1 and the control antibody at similar levels, suggesting that these cells have no or very low cell surface NY-ESO-1. We did not assess intracellular NY-ESO-1 protein, the source of NY-ESO-1 antigen presented on MHC I, in H522 cells by flow cytometry. The inability to detect the protein by Western assay could be a sensitivity issue, which we now note in the manuscript (Results, section 2.2). The NY-ESO-1 RNA was detectable in both untreated and irradiated H522 cells and was not significantly affected by radiation (Figure S4).
Unfortunately, we did not examine PD-1 by flow cytometry. However, we were able to detect the known and expected radiation-induced increase in cell surface MHC I in a few cell-lines by flow cytometry (Figure 7B).
Analyses presented were performed at four days after radiation exposure which in principle is a long period of time. Accordingly, rationales for choosing that late time point should be given and discussed in detail.
We thank the reviewer for this important aspect of our study. The duration of four days was chosen because the effect of radiation on NY-ESO-1-mediated T cell recognition was observed with it for A498 kidney [reference 17], and H522 and OE19 (Figure 1) cells. We also wanted the cells to undergo at least one doubling, because of the expected effect of radiation on cell cycle-related processes, and the largest doubling time among the seven cell-lines was 61-96 h (Table 1). We now mention these two points in Results (section 2.4). We also note this limitation of our study in Discussion (final para.).
Authors claimed to analyze mechanisms of antigen presentation, but some figures (Fig. 3, Fig. 4) mainly include assessment of proliferation and morphological characterization and describe well known phenomenons like cell cycle arrest and polyploidy in all the cell lines used. These data neither contribute novel findings nor did they include data important for the major purpose of the manuscript. Thus, these data should be removed.
As pointed out by the reviewer, our findings on cell enlargement following irradiation, likely because of cell cycle arrest, are not novel, and we note in Discussion that the cell enlargement effect has been described before in a few papers. However, the work in those papers was limited to one or two cell-lines. Our findings with seven cell-lines demonstrate the likely universality of the effect for epithelial cancer cells. Furthermore, the enlargement effect has bearing on design and interpretations of results of experiments, as we illustrate with the increased non-specific binding of cells to antibodies in flow cytometry assays that we had performed to examine the consequence of radiation on antigen presentation.
Nevertheless, we are willing to follow the reviewer’s suggestion if they consider it important.
Table 1 mainly covers published data on well-established cell lines derived from lung and head and neck malignancies but does not contain findings from the present study. Thus, the table should be moved from the results section to a supplemental file.
We agree than Table 1 does not include any new finding and is merely a summary of what is already known about the cell-lines. Because these cell-lines are the subjects of investigation in our study, a mention of their characteristics of relevance to the study, such as doubling time and radiosensitivity, within the main manuscript is important. We believe that the table format that we use is a good way to present this information. But we will accept the reviewer’s suggestion if they consider it important.
Table 2. Gene expression differences between three pairs of 15 Gy irradiated and non-irradiated cells of seven cell lines were depicted. Authors clearly should give selection criteria for these lines and reasons not to include all lines in comparisons.
In Table 2, all seven cell-lines are included. Specifically, Table 2 lists the genes that were up-regulated by radiation in all seven cell-lines. Because of space constraint, the table only shows ranges of fold-change and P values across the seven cell-lines. Fold-change and P values of each cell-line are provided in Table S1.
Figure 7B. Authors failed to display MHC expression on H522 cells to refer to the data given in figure 1, hampering to follow their statement (page 11, line 4): “… increase in gene expression for the antigen but likely because of up-regulation by radiation of HLA genes encoding for class I MHC molecules that present the antigen to T cells”.
We thank the reviewer for noting this. We did not assess cell surface MHC I protein expression in H522 cells by flow cytometry. Though MHC I gene expression was increased in H522 cells by radiation (Table 2), this observation by itself does not strongly support our statement in Discussion (para. 1) that the reviewer has pointed to.
We have now modified the statement to: “… increase in gene expression for the antigen but may be because of up-regulation of its presentation consequent to an increase in expression levels of antigen processing/presentation genes such as HLA genes encoding for class I MHC molecules that present the antigen to T cells.”
Round 2
Reviewer 2 Report
In the revised version of the manuscript, the authors addressed the previous concerns in a convincing manner or tractable argued their statements.